# The Cumulative Impacts of Fatigue during Overload Training Can Be Tracked Using Field-Based Monitoring of Running Stride Interval Correlations

**DOI:** 10.3390/s24175538

**Published:** 2024-08-27

**Authors:** Joel Thomas Fuller, Tim Leo Atherton Doyle, Eoin William Doyle, John Bradley Arnold, Jonathan David Buckley, Jodie Anne Wills, Dominic Thewlis, Clint Ronald Bellenger

**Affiliations:** 1Faculty of Medicine, Health and Human Sciences, Macquarie University, Sydney, NSW 2109, Australia; tim.doyle@mq.edu.au (T.L.A.D.); eoin.doyle@mq.edu.au (E.W.D.); jodie.wills@mq.edu.au (J.A.W.); 2Biomechanics, Physical Performance, and Exercise Research Group, Macquarie University, Sydney, NSW 2109, Australia; 3Alliance for Research in Exercise, Nutrition and Activity (ARENA), UniSA Allied Health and Human Performance Unit, University of South Australia, Adelaide, SA 5001, Australia; john.arnold@unisa.edu.au (J.B.A.); jon.buckley@unisa.edu.au (J.D.B.); clint.bellenger@unisa.edu.au (C.R.B.); 4Centre for Orthopaedic & Trauma Research, Adelaide Medical School, The University of Adelaide, Adelaide, SA 5000, Australia; dominic.thewlis@adelaide.edu.au

**Keywords:** athlete performance, accelerometry, inertial measurement unit, movement variability, detrended fluctuation analysis, training load

## Abstract

Integrating running gait coordination assessment into athlete monitoring systems could provide unique insight into training tolerance and fatigue-related gait alterations. This study investigated the impact of an overload training intervention and recovery on running gait coordination assessed by field-based self-testing. Fifteen trained distance runners were recruited to perform 1-week of light training (baseline), 2 weeks of heavy training (high intensity, duration, and frequency) designed to overload participants, and a 10-day light taper to allow recovery and adaptation. Field-based running assessments using ankle accelerometry and online short recovery and stress scale (SRSS) surveys were completed daily. Running performance was assessed after each training phase using a maximal effort multi-stage running test-to-exhaustion (RTE). Gait coordination was assessed using detrended fluctuation analysis (DFA) of a stride interval time series. Two participants withdrew during baseline training due to changed personal circumstances. Four participants withdrew during heavy training due to injury. The remaining nine participants completed heavy training and were included in the final analysis. Heavy training reduced DFA values (standardised mean difference (SMD) = −1.44 ± 0.90; *p* = 0.004), recovery (SMD = −1.83 ± 0.82; *p* less than 0.001), performance (SMD = −0.36 ± 0.32; *p* = 0.03), and increased stress (SMD = 1.78 ± 0.94; *p* = 0.001) compared to baseline. DFA values (*p* = 0.73), recovery (*p* = 0.77), and stress (*p* = 0.73) returned to baseline levels after tapering while performance trended towards improvement from baseline (SMD = 0.28 ± 0.37; *p* = 0.13). Reduced DFA values were associated with reduced performance (r^2^ = 0.55) and recovery (r^2^ = 0.55) and increased stress (r^2^ = 0.62). Field-based testing of running gait coordination is a promising method of monitoring training tolerance in running athletes during overload training.

## 1. Introduction

Athlete monitoring is a critical practice in high-performance sports to optimise athlete training and preparedness for competition as well as support injury and illness prevention [1,2]. Ongoing sports science research seeks to optimise the assessments used to monitor athletes and the interpretation of athlete monitoring data [2]. Optimal monitoring of athletes during times of intense training should (1) allow identification of athletes who are pushed too far into a fatigued state, (2) enable early intervention to restore desired performance capabilities and avoid adverse events, and (3) provide insight into the readiness of athletes to return to intense training following recovery interventions. Various assessment approaches are considered relevant to effectively track athletes across the fatigue-fitness adaptation continuum. To date, most assessments used for this purpose include GPS and heart rate (HR) training metrics, self-reported measures of psychological health, and various physiological and biochemical parameters [3].

Assessing motor system output represents an underexplored area within athlete monitoring relative to other assessment approaches and may provide unique insights into athlete training states that complement existing options. Fatigue and motor system output are known to interact through peripheral and central mechanisms [4,5]. However, recent investigations of current athlete monitoring practices did not identify motor system-based approaches amongst practitioner reports [1,3]. This likely relates to a lack of well-established methods for using these approaches in athlete monitoring systems. Additionally, GPS, HR, and questionnaire data are more accessible to practitioners compared to 3D kinematic and kinetic data, which often require expensive and technical laboratory equipment [6]. Recent enhancements in the wearable sensors used for movement analysis may help to overcome the latter barrier and help establish useful approaches for routine monitoring of motor system output in athletes [6]. Integrating this data with information from GPS, HR, physiological, biochemical, and self-reported health assessments could further broaden the understanding of how athletes respond to training and improve monitoring precision.

Running is an activity that has been explored extensively in motor system research and is also a fundamental component of training and competition for many athletes. Implementing motor system output assessments within the running activity of athletes could provide a feasible and valuable strategy for attaining motor system-based athlete monitoring insights for athletes who run regularly. Early studies exploring stride-based markers of gait coordination have demonstrated that detrended fluctuation analysis (DFA) of stride interval data can detect alterations caused by acute fatigue [7], functional overreaching [8,9], and recent injury [7]. Stride interval is only one of many gait parameters that can be considered; however, it is a convenient measure for field testing because it requires only a single sensor and minimal data processing compared to other parameters (e.g., joint kinematics and kinetics). DFA is a valuable method for analysing stride interval time series data because it detects complex non-linear patterns in running stride sequencing that other methods that focus on stride interval mean and SD cannot [7,8]. DFA determines the degree of long-range correlations in time series gait data. Long-range correlations in stride interval are thought to be regulated by supraspinal control mechanisms within the central nervous system (CNS) [10], which may be sensitive to the impacts of fatigue [8]. The presence of strong long-range correlations in gait is considered reflective of healthy locomotor function in contrast to random stride dynamics, which can reflect fatigue and injury [7] as well as poor health and function [11]. During longitudinal monitoring of athletes undertaking intense training, stride interval correlation reduces when athletes are functionally overreached, and this is moderately correlated with increased athlete-reported training and life stressors and declining physical performance [8,9]. These moderate correlations may reflect the potential for CNS fatigue mechanisms to have overlapping negative impacts on gait regulation, physical performance, and athlete mood.

Existing research into stride interval correlation behaviour during fatigue conditions has often focused on responses to a single prolonged run [7,12] rather than repeated exposure to intense training. Longitudinal study designs reflect athlete monitoring in practical settings (i.e., repeated athlete observations over time) and provide ecologically valid insights into the application of stride interval correlation assessments. Notably, the studies investigating stride interval correlation behaviour during repeated intense training (i.e., overload training leading to functional overreaching) have collected data in laboratory settings [8,9]. Relying on laboratory assessments limits the generalizability of findings and can prevent research from being translated effectively to the field settings required in high-performance sports. It also limits testing frequency and scale based on laboratory access and size. A recent field-based study demonstrated promising feasibility for monitoring stride interval correlations using athlete self-testing with wearable accelerometers during 1-week of training [13]. This method could be translated to high-performance sports settings where regular running training is undertaken and careful monitoring of fatigue is required. Establishing fatigue markers specific to running gait patterns may be particularly relevant to running-focused athletes who are often at increased risk of overuse running injuries when aberrant biomechanics are present [14]. However, the proposed field-based self-testing approaches must be sensitive to the accumulation of fatigue during overload training in the same manner as laboratory testing.

We aimed to investigate the impact of a 1-month running training intervention consisting of light, heavy, and taper training phases on stride interval correlations assessed by field-based self-testing. Additionally, we investigated whether changes in field-based stride interval correlation measurements were associated with changes in running performance and self-reported performance and stress. We hypothesised that stride interval correlations would be reduced after heavy training compared to baseline and taper training, and these changes would be linearly associated with changes in running performance, self-reported performance, and stress.

## 2. Materials and Methods

### 2.1. Participants

This observational study was approved by the Macquarie University Human Research Ethics Committee. Fifteen distance runners provided written informed consent to participate and were enrolled in the study. Runners were eligible if they were aged 18–44 years, running ≥20 km/week for at least the previous 6 months, and capable of running 5 km in ≤23 min. Runners were excluded if they had a current or recent (within the previous 3 months) running injury. Sample size estimation in G*Power (version 3.1.9.4) [15] determined that a minimum of nine participants were required to detect the expected large changes (f = 0.50) associated with heavy training across the 3 measurement timepoints with 80% power and 0.05 alpha. This aligned with previous studies investigating changes in similar health outcome measures during heavy training [9,16].

### 2.2. Study Design

The study design involved repeated measures during exposure to three different training blocks in a fixed order. The order was 1 week of light (baseline) training, 2 weeks of overreaching (heavy) training, and 1.5 weeks of taper training. Field-based assessment of stride intervals and questionnaire assessment of training tolerance were completed with each training session or daily. A running performance assessment was completed 1-day after the final training session of each training block. Participants completed two practice sessions before starting the study to reduce the impact of learning effects during the observation period. This included practising field testing away from the laboratory environment. The time of day was standardised for the performance testing of each participant. Participants were instructed not to complete training on the day of performance testing and to standardise meals and hydration before tests.

### 2.3. Training Intervention

The training program was similar to previous studies in this area and was based on prescription at relative intensities (% heart rate maximum [HRmax]) [16]. All training was pre-programmed and facilitated using a Garmin watch (Forerunner 235, Garmin International, Olathe KS, USA) with an HR monitor chest strap (HRM-Dual, Garmin International, Olathe, KS, USA). Light training involved training on 6 of 7 days, with each session involving running 30 min at 65–75% HRmax. Heavy training involved training on 14 of 14 days using 4 sets of 2–4-min intervals described previously [9]. The interval zones (duration) were 69–81% (4 min), 82–87% (4 min), 88–94% (4 min), and >94% (2 min) HRmax. Taper training involved training on 7 of 9 days with 5x 30-min sessions at 65–75% HRmax, 1x 25-min session at 75–85% HRmax, and 1x interval session involving 4 sets of 69–81% (3 min) and 88–94% (2 min) zones.

### 2.4. Field Testing

Stride interval field testing used methods described previously [13]. One Shimmer3 inertial measurement unit (IMU; Shimmer, Dublin, Ireland) was attached with a Velcro strap to the distal medial tibia with the y-axis (vertical) aligned to the long axis of the tibia. Sensor location was standardized to the right leg for all assessments. A single sensor was used to minimise participant burden due to between-leg symmetry not being a focus of this study. Axial acceleration was recorded at 504-Hz to the onboard memory while participants completed a 6-min warm-up run at 60–70% HRmax immediately before each training session. The Garmin watch provided real-time feedback to help participants keep within the prescribed HR, which was chosen based on previous research into optimal relative intensity for this testing [9]. Participants attached the IMUs and started and stopped recordings for themselves after receiving training and practising this during the practice period (before starting the observation period). They were instructed to complete the testing on a level running track near their area of residence and standardise the test location throughout the study. Additionally, they were instructed to maintain their usual pre- and post-running routine (i.e., food intake, hydration, and stretching). Data were exported at the end of each training block. Garmin watch GPS data were reviewed to compare running locations used by participants.

### 2.5. Short Recovery and Stress Scale

Self-reported recovery and stress were assessed daily using an online version of the 8-item short recovery and stress scale (SRSS) [17]. It consists of eight items on a 7-point scale ranging from 0 (does not apply at all) to 6 (fully applies). The eight items included (1) Physical Performance Capability, (2) Mental Performance Capability, (3) Emotional Balance and (4) Overall Recovery as measures of recovery, (5) Muscular Stress, (6) Lack of Activation, and (7) Negative Emotional State and (8) Overall Stress as measures of stress. High scores indicated improved recovery for the recovery items (item 1-4) and high stress for the stress items (item 5-8). Scores for Overall Recovery (item 4) and Overall Stress (item 8) were the specific outcomes used for this study. The Overall Recovery item asked participants to “rate how you feel right now in relation to your best ever overall recovery state (e.g., recovered, rested, muscle relaxation, physically relaxed)”. The Overall Stress item asked participants to “rate how you feel right now in relation to your highest ever overall stress (e.g., tired, worn-out, overloaded, physically exhausted)”. The average daily score for both the Overall Recovery and Overall Stress items across the final 7 days in each training phase was used for statistical analysis.

### 2.6. Running Performance

Running performance was assessed with a maximal effort multi-stage running test to exhaustion (RTE) on a motorised treadmill set at 0% grade (AMTI, Watertown, MA, USA). Participants completed a 6-min warm-up run at their preferred running speed before starting the test. The multi-stage test consisted of consecutive 5-min running stages at progressively increasing running speeds. The first stage was set at a running speed of 10 km/h (2.78 m/s). Each subsequent 5-min stage was set at a running speed that was 1 km/h (0.28 m/s) faster than the previous stage. Participants continued running throughout the progressively faster stages until volitional exhaustion was reached and they were unable to keep running despite standardized encouragement from the tester. The total cumulative running distance covered from starting the test to reaching volitional exhaustion was the performance measure. The greater running distance indicated greater running performance (i.e., greater running distance covered before reaching a state of exhaustion). Participants reported their perceived exertion after completing the test using a 6-to-20 rating of perceived exertion (RPE) scale [18]. Scores of 20 reflected maximal perceived exertion. The RTE protocol was adapted from a previous study evaluating endurance athletes during functional overreaching [19]. Analysis of test–retest data from 12 runners completing this test in our laboratory indicated a high intra-class correlation (ICC = 0.98) and a 0.29 km (6.2%) typical error. Natural logarithmic transformation of total distance data was required to achieve a normal distribution prior to statistical analysis.

### 2.7. IMU Data Processing

IMU data were processed in Matlab (version R2023a). A low-pass fourth-order Butterworth filter was applied using an 80-Hz cut-off selected from residual analysis. Separate ground contacts were identified from the tibial long axis (vertical) acceleration using the *findpeaks* function and stride intervals were calculated as the time between consecutive peaks. The initial 60 s of data were excluded due to potential start-up effects on running gait (e.g., speed fluctuations before HR achieves a steady state). The subsequent 300 strides were exported as the stride interval time series for analysis. These steps were based on previous approaches [8,9].

DFA was performed on time series data using PhysioNet software (version 1.0.0) [20] to determine the long-range correlation coefficient alpha (α). The PhysioNet DFA algorithm integrated the time series and then sectioned it into data bins of length N. The range for N was a minimum of 4 strides to a maximum of 75 strides. Next, a least square trend line was used to fit each data bin. The resulting trends were subtracted from each respective bin to detrend the data. The root mean square (RMS) fluctuation in the integrated and detrended time series was determined for each bin length. The coefficient (DFA α) relating RMS magnitude to bin length on a log-log plot represented the degree of long-range correlation present in the stride interval time series; a 1.00 value indicated perfect correlation and 0.50 indicated random noise [21]. Two-day moving average DFA α values were used for analysis based on improved smallest detectable change capability when compared to single-day values [13].

### 2.8. Statistical Analysis

SPSS (v28, IBM, New York, NY, USA) was used for statistical analysis. The normality of data was investigated using Shapiro–Wilk tests. DFA α, running performance, recovery, and stress scale data were compared across the training intervention using data from the end of each training phase (post-light [baseline], post-heavy, and post-taper). Comparisons were based on mixed-effect models that included the training phase as a fixed effect with repeated measures and participants as a random effect. All participants completing heavy training were included in the models. Post-hoc pairwise comparisons were made using Fisher’s least significant difference tests. Statistical significance was set at alpha equals 0.05. The standardised mean difference (SMD) was used for the effect size and was considered small if <0.50, moderate if 0.50–0.79, and large if ≥0.80 [22]. Effect precision was based on the 95% confidence interval (CI). Outcomes with a non-normal distribution were analysed using the Friedman test.

Analysis of covariance (ANCOVA) was used to determine the within-subject correlation between change scores for each outcome measure. Change scores using DFA α, running performance, recovery, and stress scale data were calculated for each pairwise contrast (heavy vs. baseline, taper vs. baseline, and taper vs. heavy). The ANCOVA included DFA change as a covariate and either running performance, recovery, or stress scale changes as the dependent variable. The participant was considered a fixed effect to account for the repeated measures on each participant. The correlation coefficient (r) was used for the effect size and was considered small if <0.30, moderate if 0.30–0.49, and large if ≥0.50 [23].

## 3. Results

### 3.1. Participant Flow, Demographics, and Compliance

Fifteen (n = 15) participants were enrolled and commenced the study. Two participants withdrew during baseline training due to changes in personal circumstances that prevented them from continuing. Four participants withdrew during heavy training due to lower limb injury (three participants) or exercise-related chest pain (one participant). The remaining nine participants (eight male and one female) were included in the final analysis. The mean (SD) demographic information for these participants were the following: age 37.8 (4.0) years; height 1.73 (0.06) m; mass 70.8 (7.9) kg; weekly distance 44 (21) km; and 5 km time: 21:01 (1:37) minutes:seconds. One participant had missing data for the post-taper timepoint due to sickness that prevented them from completing that training and testing phase.

Compliance with training was excellent (98% of overall training sessions completed). Across training blocks, there was one total missed light [baseline] training session, two total missed heavy training sessions, and two total missed taper training sessions. Participants failed to record IMU field test data for 1.8% of sessions, and 2.3% of recordings were not useable due to the lack of identifiable peaks. The remaining 95.9% of total IMU field tests were successfully recorded and used in analysis. Mean field test %HRmax was 71.5 ± 4.1% and was not different between light [baseline], heavy, or taper training (main effect *p* = 0.34; mean difference: 0.2–1.4% between phases). One participant completed all field testing on a level grass sports field. Two participants completed field testing on either a level grass sports field or a flat pavement/asphalt surface. All other participants completed all field testing on a flat pavement/asphalt surface.

### 3.2. DFA α Outcomes

DFA α did not show evidence of non-normality (W = 0.975, *p* = 0.76). There was a significant main effect of the training phase on DFA α (Table 1). DFA α demonstrated a large decrease at the end of heavy vs. light [baseline] training (SMD = −1.44 ±0.90; *p* = 0.004) and no difference at the end of taper vs. light [baseline] training (SMD = −0.16 ±0.96; *p* = 0.73). Two-day moving average DFA α values from the entire observation period are presented in Figure 1. Values reduced during the final two days of heavy training and then increased gradually back towards the baseline across the first five taper sessions.

### 3.3. Questionnaire and Performance Outcomes

RTE performance did not show evidence of non-normality when analysed on the natural logarithmic scale (W = 0.960, *p* = 0.38). There was a significant main effect of the training phase on RTE performance (Table 1). Performance decreased following heavy vs. light training (SMD = −0.36 ± 0.32; *p* = 0.03) and then demonstrated a non-significant increase following taper vs. light training (SMD = 0.28 ± 0.37; *p* = 0.13). The back-transformed pairwise differences were −12.1% (95% CI: −1.4, −21.7%) and 10.9% (95% CI: −3.4, 27.1%), respectively. RPE reported immediately post-RTE showed evidence of non-normality (W = 0.742, *p* < 0.001). The Friedman test indicated that RPE immediately post-RTE was not different across training phases (χ^2^ = 0.67; *p* = 0.72; Table 1).

SRSS overall stress (W = 0.967, *p* = 0.55) and recovery (W = 0.971, *p* = 0.66) ratings did not show evidence of non-normality. There was a significant main effect of the training phase on overall stress and recovery (Table 1). Reported recovery scores demonstrated a large decrease following heavy vs. light training (SMD = −1.83 ± 0.82; *p* < 0.001) and no difference following taper vs. light training (SMD = 0.10 ± 0.76; *p* = 0.77). Reported stress scores demonstrated a large increase following heavy vs. light training (SMD = 1.76 ± 0.94; *p* = 0.001) and no difference following taper vs. light training (SMD = −0.16 ± 0.98; *p* = 0.73).

### 3.4. Correlation Analysis

There were large within-subject correlations between the change in DFA α and the changes in RTE performance (r = 0.74), self-reported recovery (r = 0.74), and self-reported stress (r = −0.79) (*p* < 0.01 for all; Figure 2). Reduction in DFA α was associated with reduced distance covered in RTE testing, reduced reported level of recovery, and increased reported level of stress. This direction of association was uniform for 9/9, 8/9, and 8/9 participants, respectively.

## 4. Discussion

We investigated the impact of light, heavy, and taper training phases on stride interval correlations assessed by field-based self-testing and the relationships between stride interval correlation changes and other athlete monitoring outcomes. Findings demonstrated that field-based self-testing of stride interval correlation is a promising method for identifying when running gait is impacted by the fatigue accumulated during intense training. Consistent with our hypotheses, stride interval correlations were reduced after a 2-week heavy training phase, recovered by the end of a subsequent taper, and associated with changes in running performance and self-reported recovery and stress. Integrating this motor system-based athlete monitoring approach with existing GPS, HR, and health questionnaire approaches could provide a more complete picture of how fatigue impacts athletes during heavy training. This would better inform athletes and coaches on athlete readiness, thereby allowing optimisation of training prescription, and warrants further investigation in larger studies.

The large reduction in stride interval correlation after heavy training compared to baseline indicates that the accumulated fatigue caused a change in running gait coordination. The direction of this change is consistent with findings from previous heavy training interventions [9] and prolonged running observations [7]. Lower stride interval correlation values indicate that each running stride is less related to earlier strides and implies a more random pattern of stride sequencing [21]. Loss of stride interval correlation properties during gait has been previously associated with aging [24] and disease [11]. As a result, the similar reductions observed for athletes in heavily fatigued states are likely to represent a detrimental change. Indeed, these changes were associated with reduced running performance in the present study and previous similar studies [8,9]. Aberrant biomechanics are often considered a risk factor for running injury [14] and this is likely to be heightened in the context of the high training loads associated with a heavy training phase. Monitoring stride interval correlation during heavy training may help alert runners and coaches to when biomechanics have been adversely impacted by training. This could prompt coaches to reduce training intensity and/or volume to see if biomechanics subsequently return to normal or consider a more detailed assessment of joint kinematics and kinetics to understand the specific biomechanical changes that have occurred and if/how they can be addressed. Both strategies may support injury prevention during heavy running training. Future research with large runner cohorts is required to determine stride interval correlation cut-offs for detecting when biomechanics have been meaningfully impacted to a degree that requires actions to be taken.

Taper training was accompanied by the recovery of stride interval correlation properties back to baseline values, which was associated with increased running performance. This contrasts with a similar study that found that reductions in stride interval correlation after heavy training were still present after a subsequent taper [9]. Both studies used the same training intervention. However, the previous study [9] performed stride interval testing on a treadmill as opposed to the field-based self-testing approach used in the present study. We speculate that field running with two-day moving average values may be more sensitive to stride interval correlation changes compared to treadmill running due to the inherent constraints on running strides that result from being on a treadmill. Indeed, previous studies have observed differences in gait variability patterns between treadmill and outdoor running [25,26]. Our field-testing approach resulted in slightly higher testing HRs (72% HRmax) compared to our 65% HRmax prescription based on the previous treadmill testing study [9]. This increased value likely relates to environmental impacts on running HR [27] and suggests that runners and coaches wishing to use the field-testing approach should adopt a slightly higher HR intensity prescription (e.g., 70–75% HRmax).

Athlete self-testing of stride interval correlations was successfully performed in field environments and could be integrated into existing athlete monitoring systems to provide unique insight. Only 4% of field test accelerometry data were not collected successfully despite the athletes performing over 200 total tests during the study, indicating the robustness of this approach. The stride interval correlation values resulting from the tests aligned with expectations based on previous studies that performed testing in highly controlled and supervised settings [21,28]. Accelerometers are present in many existing sensors used routinely with athletes and may support the integration of stride interval assessment. Correlations between the changes in stride interval correlation, running performance, and self-reported stress and recovery were strong but not perfect. Equivalent coefficient of determination values (r^2^ = 0.55–0.62) indicated that 38–45% of variance remained unexplained. This unexplained variance demonstrates the degree of unique influences that each fatigue marker may be responding to and highlights the value of a multimodal measurement approach for athlete monitoring.

The present study has limitations that should be considered when interpreting and applying the findings. First, the small sample size available after participants dropped out during baseline and heavy training provided statistical power for the detection of large effects but did not represent a diverse sample of runners (e.g., varied runner sex and running ability). This limits the generalisability of the study findings, particularly for female runners, given there was only one female runner in the study cohort. A carefully curated cohort with a larger sample matching population-level distributions would make the results more generalisable. Second, the methods required regular running activity by participants. This limits the relevance of the methods for athletes who run infrequently and means they cannot be applied to other common modes of training. Third, the field-based self-testing methods employed in this study were inherently less finely controlled compared to the usual laboratory-based researcher-supervised testing. Participants were able to complete their field-based testing and training at the most convenient times for them and weather inherently varied across the 1-month observation period for each participant. These factors are expected to have added variance to our field data. However, our field-based approach was more ecologically valid than laboratory-based alternatives, we experienced minimal data loss, and our results aligned with what was expected based on laboratory studies. This represents an important step forward from laboratory findings to real-world application.

## 5. Conclusions

With appropriate educational material, stride interval correlations could be self-monitored by runners using a field-testing approach that detected alterations to running gait caused by prolonged heavy training. These changes were recovered after taper training and were associated with running performance and self-reported measures of performance and stress. Integrating stride interval correlation testing into existing athlete monitoring systems may provide unique insight into how training fatigue impacts gait and could help optimise training prescription for athlete performance and safety.

## Figures and Tables

**Figure 1 sensors-24-05538-f001:**
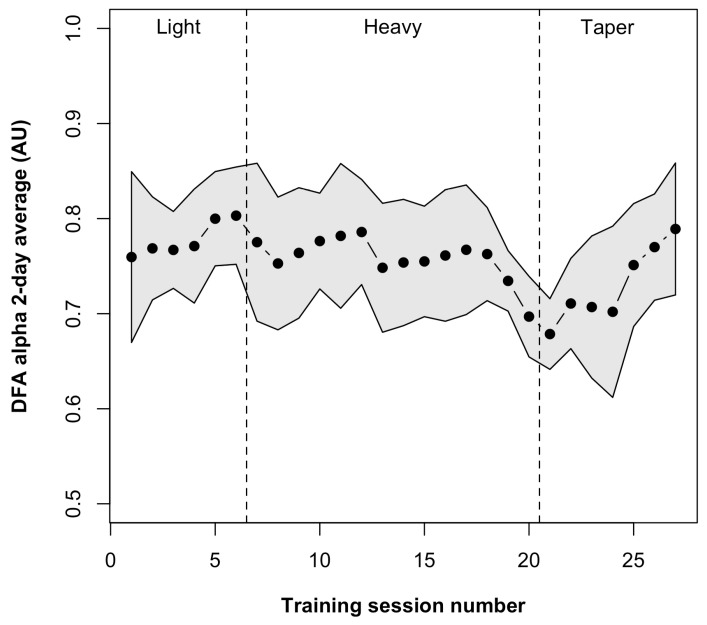
Two-day moving average DFA α values across light (baseline), heavy, and taper training phases. Values are mean two-day moving average with the shaded region indicating the 95% confidence interval. AU, arbitrary units; DFA, detrended fluctuation analysis.

**Figure 2 sensors-24-05538-f002:**
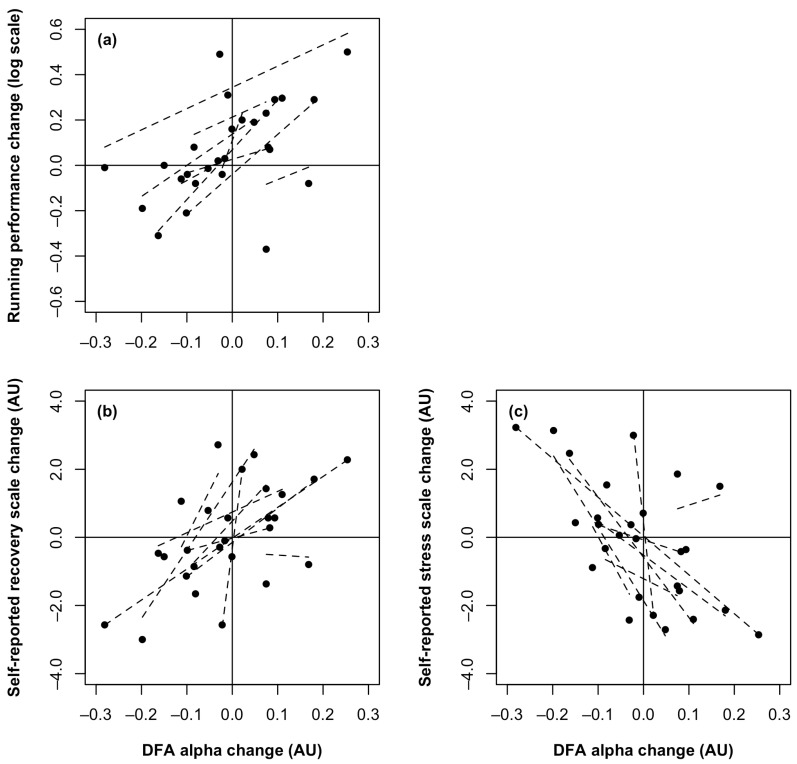
Relationship between changes in detrended fluctuation analysis (DFA) α with changes in run-to-exhaustion (RTE) performance (**a**), Short Recovery and Stress Scale (SRSS) overall recovery scores (**b**), and overall stress scores (**c**). Change scores were calculated for each participant for each pairwise comparison (heavy minus baseline, taper minus baseline, and taper minus heavy). Positive values on the y-axis indicate improved running performance (increased RTE distance), greater self-reported recovery (higher SRSS recovery score), and greater self-reported stress (higher SRSS stress score). Positive values on the x-axis indicate increased DFA α. Dashed lines represent the linear trendline for each participant. AU, arbitrary units.

**Table 1 sensors-24-05538-t001:** Study outcome measures at each assessment timepoint. AU, arbitrary units; DFA, detrended fluctuation analysis; RPE, rating of perceived exertion; RTE, run-to-exhaustion; SD, standard deviation; SRSS, Short Recovery and Stress Scale. Higher SRSS overall recovery scores indicate greater recovery. Higher SRSS overall stress scores indicate greater stress. Values are mixed model estimated marginal means.

Outcome	Post-Baseline Mean (SD) *	Post-Heavy Mean (SD) *	Post-Taper Mean (SD) *	*p*-Value Main Effect
DFA α (AU)	0.80 (0.08)	0.70 (0.07) ^1^	0.79 (0.10) ^2^	0.011
RTE distance (log scale)	1.705 (0.369)	1.576 (0.357) ^1^	1.808 (0.375) ^2^	0.007
RTE RPE (6–20 scale)	20 (19–20)	19 (19–20)	20 (19–20)	0.717
SRSS recovery (0–6 scale)	4.5 (0.8)	2.9 (0.9) ^1^	4.5 (0.7) ^2^	<0.001
SRSS stress (0–6 scale)	1.8 (1.1)	3.6 (0.9) ^1^	1.7 (0.6) ^2^	<0.001

* RTE RPE values are reported as median (25th–75th percentile) due to the non-normal distribution. ^1^ Significant pairwise difference from post-baseline (*p* < 0.05). ^2^ Significant pairwise difference from post-heavy (*p* < 0.05).

## Data Availability

Data will be available upon reasonable request where ethical approval for secondary usage has been attained.

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
