# Peer review of "The Cumulative Impacts of Fatigue during Overload Training Can Be Tracked Using Field-Based Monitoring of Running Stride Interval Correlations"

_sensors, 2024, doi:10.3390/s24175538_

Round 1

Reviewer 1 Report

Comments and Suggestions for Authors

The authors present a very interesting study where they investigated the sensitivity of detrended fluctuation analysis of stride interval data in tracking responses to different training intensities in a group of distance runners. The authors moved the training from in the laboratory to in the “wild” in order to enhance the ecological validity of the findings. The authors present a clear rationale for the study in the Introduction. Unfortunately, they experienced almost 50% drop-out from the original sample (it is not clear if 6 or 7 participants withdrew/presented incomplete data), some of which was caused by the imposed training. While the authors are clear about this in the Methods section, the Abstract needs to be revised to reflect this. Furthermore, the authors need to present an analysis of the data to ensure that the data is normally distributed. Only when this has been established can the reader be certain that the appropriate statistics have been used.

The authors need to declare whether there were any methods of ensuring that the participants performed the runs during the two-week “heavy” training block on the same course and at a similar time of day and also if there were any methods employed to record the weather conditions each time (i.e. use of participant reports). These represent potentially confounding variables.

I would also like the authors to make the following revisions:

Abstract, line 18: Include “distance runners” so that the reader is aware of the sample characteristics. Also include this term in the description of the participants in line 109.

Introduction, line 98: Change to “We aimed to investigate...”

Introduction, line 100: Change to “Additionally, we investigated...”

Discussion, line 284: Begin the Discussion by restating the purpose.

Author Response

Comment #1:The authors present a very interesting study where they investigated the sensitivity of detrended fluctuation analysis of stride interval data in tracking responses to different training intensities in a group of distance runners. The authors moved the training from in the laboratory to in the “wild” in order to enhance the ecological validity of the findings. The authors present a clear rationale for the study in the Introduction.”

Response #1: Thank you for taking time to review our manuscript and provide feedback. We appreciate your positive feedback and constructive comments, which have helped to improve our work. We have provided responses to each of your specific comments below. Edited or revised sentences in the resubmitted manuscript are colored red to help identify them.

Comment #2: “Unfortunately, they experienced almost 50% drop-out from the original sample (it is not clear if 6 or 7 participants withdrew/presented incomplete data), some of which was caused by the imposed training. While the authors are clear about this in the Methods section, the Abstract needs to be revised to reflect this.

Response #2: We are glad the Methods section presented this clearly. We have revised the Abstract to make the dropout clearer in that section. The revised text reads (Line #24-27):

“Two participants withdrew during baseline training due to changed personal circumstances. Four participants withdrew during heavy training due to injury. The remaining nine participants completed heavy training and were included in the final analysis.”

Comment #3: “Furthermore, the authors need to present an analysis of the data to ensure that the data is normally distributed. Only when this has been established can the reader be certain that the appropriate statistics have been used.

Response #3: This analysis has been added to the Results section, with accompanying information also added to the Methods section. This did not change the results in any way that impacts the Discussion section so there were no changes made to the Discussion in relation to the addition of these analyses.

The additions to the Methods section read:

“Normality of data were investigated using Shapiro–Wilk tests.” (Line #229-230)

“Outcomes with a non-normal distribution were analysed using the Friedman test.” (Line #239-240)

The additions to the Results section read:

“DFA α did not show evidence of non-normality (W=0.975, p=0.76).” (Line #272)

“RTE performance did not show evidence of non-normality when analysed on the natural logarithmic scale (W=0.960, p=0.38).” (Line #292-293)

“RPE reported immediately post-RTE showed evidence of non-normality (W=0.742, p<0.001). The Friedman test indicated that RPE immediately post-RTE was not different across training phases (X2=0.67; p=0.72; Table 1).” (Line #298-300)

“SRSS overall stress (W=0.967, p=0.55) and recovery (W=0.971, p=0.66) ratings did not show evidence of non-normality.” (Line #301-302)

Values in Table 1 were updated for RPE to report median with 25th and 75th percentile values.

Comment #4: “The authors need to declare whether there were any methods of ensuring that the participants performed the runs during the two-week “heavy” training block on the same course and at a similar time of day and also if there were any methods employed to record the weather conditions each time (i.e. use of participant reports). These represent potentially confounding variables.

Response #4: Participants did not keep a record of weather conditions during the study. We agree that variable weather conditions can influence running tests. This is an inherent limitation of field research, particularly in a training intervention such as the intervention in our study that requires consecutive training sessions across multiple weeks and cannot postpone sessions based on weather due to the impact that taking days off has on weekly training load. We have added this to the limitations acknowledged in the Discussion section. The added text is included following the next paragraph.

All performance testing in the research lab (i.e., maximal effort multi-stage running test to exhaustion) was performed at a standardised time of day for each participant because time of day can impact physical performance. Participants were not required to standardise the time of day of their daily field-testing warm-up run and training sessions due to the high burden this would place on participants who needed to integrate study commitments into their working week and daily life for the 4.5-week duration of the study. We are not aware of existing research that suggests time of day influences stride interval correlation strength. However, we recognise that is a possibility and have added this to the limitations acknowledged in the Discussion section. The added text acknowledging this limitation, and the limitation discussed in the previous paragraph is copied here:

“Participants were able to complete their field-based testing and training at the most convenient times for them and weather inherently varied across the 1-month observation period for each participant. These factors are expected to have added variance to our field data.” (Line #399-402)

Review of Garmin watch GPS information from participants indicated that participants completed their field testing at locations with a standardised and level running surface type. This information has been added to the Methods and Results sections. The added text reads:

“Garmin watch GPS data was reviewed to compare running locations used by participants.” (Line #169-170)

“One participant completed all field testing on a level grass sports field. Two participants completed field testing on either a level grass sports field or flat pavement/asphalt surface. All other participants completed all field testing on a flat pavement/asphalt surface.” (Line #267-270)  

Comment #5: “Abstract, line 18: Include “distance runners” so that the reader is aware of the sample characteristics. Also include this term in the description of the participants in line 109.

Response #5: This revision has been made at both manuscript locations as recommended (Line #18 and #120 in the revised manuscript).

Comment #6: “Introduction, line 98: Change to “We aimed to investigate...”

Response #6: This revision has been made to the Introduction (Line #109 in the revised manuscript).

Comment #7: “Introduction, line 100: Change to “Additionally, we investigated...”

Response #7: This revision has been made to the Introduction (Line #111 in the revised manuscript).

Comment #8: “Discussion, line 284: Begin the Discussion by restating the purpose.

Response #8: The beginning of the Discussion has been revised so that the purpose is restated (Line #325-327 in the revised manuscript).

Reviewer 2 Report

Comments and Suggestions for Authors

Review for The cumulative impacts of fatigue during overload training can be tracked using field-based monitoring of running stride interval correlations

The introduction is well written the only point I would like to bring up is that some athletes don’t run as regular within their training so this method could be limited to athletes focused on running performance, such as marathon and sprinting.

For the attachment of the IMU sensor why was only 1 sensor used? Would it have been nice to be able to compare both sides. Also, which side was chosen was it the dominant or non-dominant leg?

In the field-testing section did the athletes do some stretching prior or post workout? It would be prudent to include details of athletes’ preparation prior treatments.

Could the DFA be used to suggest a cut off for the athletes to show that they are training too hard?

From a practical point of few of the trainer or athlete how could this help them control the intensity of their training?

Author Response

Comment #1: “The introduction is well written the only point I would like to bring up is that some athletes don’t run as regular within their training so this method could be limited to athletes focused on running performance, such as marathon and sprinting.

Response #1: Thank you for taking time to review our manuscript and provide feedback. We appreciate your positive feedback and constructive comments, which have helped to improve our work. We have revised the Introduction to incorporate the point you have raised here. All edited or revised sentences in the resubmitted manuscript are colored red to help identify them. The specific revised content of the Introduction in relation to this comment reads (Line #67-70):

“Implementing motor system output assessments within the running activity of athletes could provide a feasible and valuable strategy for attaining motor system-based athlete monitoring insights for athletes who run regularly.”

This has also been noted as a limitation in the Discussion section in connection with our response to Reviewer #3, Comment #5. The added text reads (Line #395-397):

“The methods required regular running activity by participants. This limits the relevance of the methods for athletes who run infrequently and means they cannot be applied to other common modes of training.”

Comment #2: “For the attachment of the IMU sensor why was only 1 sensor used? Would it have been nice to be able to compare both sides. Also, which side was chosen was it the dominant or non-dominant leg?

Response #2: We chose to use one sensor only due to practical reasons related to number of available sensors for the project and to reduce the demands on participants (i.e., minimize the number of sensors they needed to manage throughout the study period). Investigating symmetry between sides is an interesting avenue of research but was not an aim of this project. We standardised sensor location to the right leg for all participants and assessments.

A summary of the above has been added to the Methods section. The added text reads (Line #157-159):

“Sensor location was standardized to the right leg for all assessments. A single sensor was used to minimise participant burden and due to between-leg symmetry not being a focus of this study.”

Comment #3: “In the field-testing section did the athletes do some stretching prior or post workout? It would be prudent to include details of athletes’ preparation prior treatments.

Response #3: Athletes were instructed to maintain their usual pre- and post-running routine, including food intake, hydration, and stretching. This was considered a more appropriate approach compared to prescribing a routine that may have been different from their usual practices and inadvertently impacted our study outcomes. Details of these instructions have been added to the Methods section. The added text reads (Line #167-168):

“Additionally, they were instructed to maintain their usual pre- and post-running routine (i.e., food intake, hydration, and stretching).”

Comment #4: “Could the DFA be used to suggest a cut off for the athletes to show that they are training too hard?

Response #4: It is possible that DFA cut offs for this purpose could be determined. However, that would require a study with a larger sample size capable of comparing the accuracy of different cut off values. We have added this as a future research direction within the Discussion section. The added text reads (Line #356-358):

“Future research with large runner cohorts is required to determine stride interval correlation cut-offs for detecting when biomechanics have been meaningfully impacted to a degree that requires action to be taken.”

Comment #5: “From a practical point of few of the trainer or athlete how could this help them control the intensity of their training?

Response #5: Practical application of this research will allow runners and coaches to monitor stride interval correlation during their training so that they can be alerted to when their biomechanics may have been adversely impacted by their training. This can be a prompt for them to reduce training intensity and/or volume to see if their biomechanics subsequently return to normal. This information has been added to the discussion in conjunction with additions from Reviewer 3, Response 4 that related to consideration of when more detailed biomechanical assessment (greater than just stride interval correlation) may be required. The section that has been revised and added to now reads as follows (Line #350-358):

“Monitoring stride interval correlation during heavy training may help alert runners and coaches to when biomechanics have been adversely impacted by training. This could prompt coaches to reduce training intensity and/or volume to see if biomechanics subsequently return to normal or consider a more detailed assessment of joint kinematics and kinetics to understand the specific biomechanical changes that have occurred and if/how they can be addressed. Both strategies may support injury prevention during heavy running training. Future research with large runner cohorts is required to determine stride interval correlation cut-offs for detecting when biomechanics have been meaningfully impacted to a degree that requires actions to be taken.”

Reviewer 3 Report

Comments and Suggestions for Authors

This study has a certain degree of innovation, but there are still several points that need improvement, as follows:

1. How to determine the sample size? 15 participants in the study, and 8 participated in the whole experiment process. Did the missing data have a great impact on the accuracy of the experiment?

2. DFA is mainly used to analyze the long-term correlation of time series data. A new level of explanation is needed as to why it is used as the main method in this experiment and the rationality of its correlation with other outcome indicators.

3. Stride is a data of spatio-temporal parameters in gait analysis. This study only involves the correlation study of stride interval to represent gait coordination. Besides the basic spatio-temporal parameters, the factors affecting human gait coordination also involve many factors such as joint Angle and torque.

4. The results of this study show that stride spacing can be used to monitor physical fatigue after high-load training, but running is only one of the conventional training methods. Is this method still sensitive and accurate for overload training other than running, and is it too single to lead to insufficient evidence?

5. The description of RTE and SRSS outcome indicators is vague, please elaborate in your response.

Author Response

Comment #1: “This study has a certain degree of innovation, but there are still several points that need improvement.

Response #1: Thank you for taking time to review our manuscript and provide feedback. We appreciate your feedback and constructive comments, which have helped to improve our work. We have revised our manuscript accordingly. The changes to the manuscript are detailed in our responses to your specific comments below. Edited or revised sentences in the resubmitted manuscript are colored red to help identify them.

Comment #2: “How to determine the sample size? 15 participants in the study, and 8 participated in the whole experiment process. Did the missing data have a great impact on the accuracy of the experiment?

Response #2: Section 2.1. described our sample size estimation. It was based on a G*Power calculation and aligned with similar previous studies (cited in the paper and listed at the end of this response). We have reviewed our description in light of your comment and added further details about the calculation to provide a more thorough description. The revised description reads (Line #124-128):

“Sample size estimation in G*Power (version 3.1.9.4) [14] determined that a minimum of nine participants were required to detect the expected large changes (f=0.50) associated with heavy training across the 3 measurement timepoints with 80% power and 0.05 alpha. This aligned with previous studies investigating changes in similar health outcome measures during heavy training [9,15].”

We described our dropout rate with reasons in Section 3.1 to ensure consistency with STROBE reporting guidelines for observational studies. Nine participants completed heavy training which was the minimum participant number required by our power calculation. The one participant we describe with missing data in the taper phase was included in our analysis due to the mixed model statistical approach we employed which can handle incomplete cases. This has been made clearer in the statistical methods description. The added text reads (Line #234-235):

“All participants completing heavy training were included in the models.”

As a result of recruiting the minimum number of participants indicated by our power calculation (after accounting for dropouts), the participants that did not complete heavy training and could not be included in the analysis did not impact our ability to detect large effects with 80% statistical power and 0.05 alpha. However, losing these participants did contribute to our study sample lacking diversity and is a factor in the limited generalisability of our study findings which we previously acknowledged. We have expanded our description of this limitation to incorporate this. The revised description of this limitation reads (Line #389-395):

“The small samples size available after participants dropped out during baseline and heavy training provided statistical power for detection of large effects but did not represent a diverse sample of runners (e.g., varied runner sex and running ability). This limits generalisability of the study findings, particularly for female runners, given there was only one female runner in the study cohort. A carefully curated cohort with a larger sample matching population-level distributions would make the results more generalisable.”

Ref 9 – Bellenger, C.R.; Arnold, J.B.; Buckley, J.D.; Thewlis, D.; Fuller, J.T. Detrended fluctuation analysis detects altered coordination of running gait in athletes following a heavy period of training. J. Sci. Med. Sport 2019, 22, 294–299. DOI: 10.1016/j.jsams.2018.09.002

Ref 14 – Faul, F.; Erdfelder, E.; Lang, A.-G.; Buchner, A. G*Power 3: A flexible statistical power analysis program for the social, be-havioral, and biomedical sciences. Behav. Res. Methods. 2007, 39, 175–191. DOI: 10.3758/bf03193146.

Ref 15 – Halson, S.L.; Bridge, M.W.; Meeusen, R.; Busschaert, B.; Gleeson, M.; Jones, D.A.; Jeukendrup, A.E. Time course of performance changes and fatigue markers during intensified training in trained cyclists. J. Appl. Physiol. 2002, 93, 947–956. DOI: 10.1152/japplphysiol.01164.2001

Comment #3: “DFA is mainly used to analyze the long-term correlation of time series data. A new level of explanation is needed as to why it is used as the main method in this experiment and the rationality of its correlation with other outcome indicators.

Response #3: As outlined in our response to your Comment #4 below, there are feasibility and practicality advantages of using stride interval as the gait parameter for our field-based experiment rather than other gait parameter options. Having selected stride interval as our gait parameter, DFA is a valuable analysis method to use because it detects complex non-linear patterns in running stride sequencing that other methods which focus on stride interval mean and SD cannot (e.g., references 7 and 8 included below). The long-range correlations in stride interval that are detected by DFA are thought to be regulated by supraspinal control mechanisms, which may be sensitive to impacts from fatigue (i.e., fatigue is also thought to have central nervous system (CNS) mechanisms). The moderate correlations between the reduced DFA alpha, reduced physical performance, and increased athlete-reported stress likely reflects CNS fatigue mechanisms causing overlapping negative impacts on gait regulation, physical performance, and athlete mood. We have integrated this rationale into the revised Introduction. The revised section reads (Line #70-89):

“Early studies exploring stride-based markers of gait coordination have demonstrated that detrended fluctuation analysis (DFA) of stride interval data can detect alterations caused by acute fatigue [7], functional overreaching [8,9], and recent injury [7]. Stride interval is only one of many gait parameters that can be considered; however, it is a convenient measure for field-testing because it requires only a single sensor and minimal data processing compared to other parameters (e.g., joint kinematics and kinetics). DFA is a valuable method for analysing stride interval time series data because it detects complex non-linear patterns in running stride sequencing that other methods which focus on stride interval mean and SD cannot [7,8]. DFA determines the degree of long-range correlations in time series gait data. Long-range correlations in stride interval are thought to be regulated by supraspinal control mechanisms within the central nervous system (CNS) [10], which may be sensitive to the impacts of fatigue [8]. The presence of strong long-range correlations in gait are considered reflective of healthy locomotor function in contrast to random stride dynamics, which can reflect fatigue and injury [7] as well as poor health and function [11]. During longitudinal monitoring of athletes undertaking intense training, stride interval correlation reduces when athletes are functionally overreached, and this is moderately correlated with increased athlete-reported training and life stressors and declining physical performance [8,9]. These moderate correlations may reflect the potential for CNS fatigue mechanisms to have overlapping negative impacts on gait regulation, physical performance, and athlete mood.”

Ref 7 – Meardon, S.A.; Hamill, J.; Derrick, T.R. Running injury and stride time variability over a prolonged run. Gait Posture 2011, 33, 36–40. DOI: 10.1016/j.gaitpost.2010.09.020

Ref 8 – Fuller, J.T.; Bellenger, C.R.; Thewlis, D., Arnold, J.; Thomson, R.L.; Tsiros, M.D.; Robertson, E.Y.; Buckley, J.D. Tracking per-formance changes with running-stride variability when athletes are functionally overreached. Int. J. Sports Physiol. Perform. 2017, 12, 357–363. DOI: 10.1123/ijspp.2015-0618

Comment #4: “Stride is a data of spatio-temporal parameters in gait analysis. This study only involves the correlation study of stride interval to represent gait coordination. Besides the basic spatio-temporal parameters, the factors affecting human gait coordination also involve many factors such as joint Angle and torque.

Response #4: We agree that there are many other options for gait analysis. However, we believe that stride interval is a convenient measure for field-testing that can be monitored using our methods more easily than joint kinematic and kinetic parameters that typically require multi-sensor systems and a greater number of computational steps. This has been made clearer in the Introduction. The revised text reads (Line #73-76):

“Stride interval is only one of many gait parameters that can be considered but it is a convenient measure for field-testing because it requires only a single sensor and minimal data processing compared to other parameters (e.g., joint kinematics and kinetics).”

Detailed biomechanical assessment remains important and can be highly complementary to our approach i.e., stride interval assessment findings may help identify the athletes in greatest need of detailed biomechanical assessment. We have edited the Discussion section to reflect this in conjunction with Reviewer #2, Comment #5 which asked about how runners and coaches could use information from our testing approach. The section that has been revised and added to now reads as follows (Line #350-358):

“Monitoring stride interval correlation during heavy training may help alert runners and coaches to when biomechanics have been adversely impacted by training. This could prompt coaches to reduce training intensity and/or volume to see if biomechanics subsequently return to normal or consider a more detailed assessment of joint kinematics and kinetics to understand the specific biomechanical changes that have occurred and if/how they can be addressed. Both strategies may support injury prevention during heavy running training. Future research with large runner cohorts is required to determine stride interval correlation cut-offs for detecting when biomechanics have been meaningfully impacted to a degree that requires actions to be taken.”

Comment #5: “The results of this study show that stride spacing can be used to monitor physical fatigue after high-load training, but running is only one of the conventional training methods. Is this method still sensitive and accurate for overload training other than running, and is it too single to lead to insufficient evidence?

Response #5: The study was aimed at running specifically so it is not possible to determine if our method will be sensitive and accurate for other training modes. We have added this as a limitation in the Discussion section in conjunction with our response to Reviewer #2, Comment #1. The added text reads (Line #395-397):

“The methods required regular running activity by participants. This limits the relevance of the methods for athletes who run infrequently and means they cannot be applied to other common modes of training.”

Comment #6: “The description of RTE and SRSS outcome indicators is vague, please elaborate in your response.

Response #6: We have added further details to the description of both of these outcomes in the Methods section to make the descriptions clearer. The revised descriptions read (Line #172-186 and Line #188-201):

“Self-reported recovery and stress was assessed daily using an online version of the 8-item short recovery and stress scale (SRSS) [16]. It consists of eight items on a 7-point scale ranging from 0 (does not apply at all) to 6 (fully applies). The eight items included (1) Physical Performance Capability, (2) Mental Performance Capability, (3) Emotional Balance and (4) Overall Recovery as measures of recovery, and (5) Muscular Stress, (6) Lack of Activation, (7) Negative Emotional State and (8) Overall Stress as measures of stress. High scores indicated improved recovery for the recovery items (item 1-4) and high stress for the stress items (item 5-8). Scores for Overall Recovery (item 4) and Overall Stress (item 8) were the specific outcomes used for this study. The Overall Recovery item asked participants to “rate how you feel right now in relation to your best ever overall recovery state (e.g. recovered, rested, muscle relaxation, physically relaxed)”. The Overall Stress item asked participants to “rate how you feel right now in relation to your highest ever overall stress (e.g. tired, worn-out, overloaded, physically exhausted)”. The average daily score for both the Overall Recovery and Overall Stress items across the final 7 days in each training phase was used for statistical analysis.”

“Running performance was assessed with a maximal effort multi-stage running test to exhaustion (RTE) on a motorised treadmill set at 0% grade (AMTI, Massachusetts, USA). Participants completed a 6-minute warm-up run at preferred running speed before starting the test. The multi-stage test consisted of consecutive 5-minute running stages at progressively increasing running speed. The first stage was set at a running speed of 10 km/h (2.78 m/s). Each subsequent 5-minute stage was set at a running speed that was 1 km/h (0.28 m/s) faster than the previous stage. Participants continued running throughout the progressively faster stages until volitional exhaustion was reached and they were unable to keep running despite standardized encouragement from the tester. The total cumulative running distance covered from starting the test to reaching volitional exhaustion was the performance measure. Greater running distance indicated greater running performance (i.e., greater running distance covered before reaching a state of exhaustion). Participants reported their perceived exertion after completing the test using a 6-to-20 rating of perceived exertion (RPE) scale [17]. Scores of 20 reflected maximal perceived exertion. The RTE protocol was adapted from a previous study evaluating endurance athletes during functional overreaching [18].”

Round 2

Reviewer 1 Report

Comments and Suggestions for Authors

The authors have completed all of the required revisions to the manuscript. The manuscript should now be accepted for publication.